# Geminiviruses employ host DNA glycosylases to subvert DNA methylation-mediated defense

Xiaojian Gui[1,2], Chang Liu[3,4], Yijun Qi [3,4 ✉] & Xueping Zhou [1,2 ✉]

DNA methylation is an epigenetic mechanism that plays important roles in gene regulation and transposon silencing. Active DNA demethylation has evolved to counterbalance DNA methylation at many endogenous loci. Here, we report that active DNA demethylation also targets viral DNAs, tomato yellow leaf curl China virus (TYLCCNV) and its satellite tomato yellow leaf curl China betasatellite (TYLCCNB), to promote their virulence. We demonstrate that the βC1 protein, encoded by TYLCCNB, interacts with a ROS1-like DNA glycosylase in *Nicotiana benthamiana* and with the DEMETER (DME) DNA glycosylase in *Arabidopsis thaliana*. The interaction between βC1 and DME facilitates the DNA glycosylase activity to decrease viral DNA methylation and promote viral virulence. These findings reveal that active DNA demethylation can be regulated by a viral protein to subvert DNA methylation-mediated defense.

[1] State Key Laboratory of Rice Biology, Institute of Biotechnology, Zhejiang University, Hangzhou 310058, China. [2] State Key Laboratory for Biology of Plant Diseases and Insect Pests, Institute of Plant Protection, Chinese Academy of Agricultural Sciences, Beijing 100193, China. [3] Center for Plant Biology, School of Life Sciences, Tsinghua University, Beijing 100084, China. [4] Tsinghua University-Peking University Joint Center for Life Sciences, School of Life Sciences, Tsinghua University, Beijing 100084, China. ✉email: qiyijun@tsinghua.edu.cn; zzhou@zju.edu.cn

C ytosine DNA methylation plays important roles in transposon silencing and gene regulation[1]. In plants, DNA methylation occurs in CG, CHG, and CHH (where H represents A, T, or C) sequence contexts[2–4]. CG, CHG, and CHH methylation are established by Domains Rearranged Methyltransferase 2 through the RNA-directed DNA methylation (RdDM) pathway[5] and maintained by different mechanisms. CG and CHG methylation are maintained by METHYLTRANSFERASE 1[6] and CHROMOMETHYLASE 3[7], respectively, whereas CHH methylation is maintained by CMT2 and RdDM[8]. All DNA methylation reactions use S-adenosyl-L-methionine (SAM) as the donor of methyl groups[9].

DNA methylation can be passively or actively removed, and the balance between methylation and demethylation determines the final methylation pattern[1]. In *Arabidopsis*, active DNA demethylation involves the removal of 5-methylcytosine (5mC) by the DME family of DNA glycosylases, which include REPRESSOR OF SILENCING 1 (ROS1)[10], DEMETER (DME)[11], DEMETER-LIKE PROTEIN 2 (DML2), and DML3[12]. ROS1, DML2, and DML3 are expressed in vegetative tissues, whereas DME is preferentially expressed in the gametes[11,13]. ROS1, DML2, and DML3 target thousands of endogenous loci[2,4,14] and are important for the development of xylem tracheary elements and leaf stomatal cells[15,16]. DME targets thousands of endogenous loci in vegetative cells, the companion cells of sperms[17], and initiates genome-wide active DNA demethylation in central cells, the companion cells of egg cells[18], thereby regulating pollen germination[13] and embryo development[11].

Plant viruses infect a wide range of plant species and cause devastating crop losses all over the world. The family *Geminiviridae* comprises circular single-stranded DNA viruses. This family is currently divided into 14 genera on the basis of their genome organization, insect vector, and host range, with the genus *Begomovirus* being the largest[19]. The begomovirus tomato yellow leaf curl China virus (TYLCCNV), which can infect *Nicotiana benthamiana*, is associated with a betasatellite DNA (TYLCCNB)[20]. TYLCCNB encodes βC1, which determines the pathogenicity of TYLCCNV[20]. The multifunctional protein βC1 can block Mitogen-Activated Protein Kinase signaling[21], interfere with post-transcriptional and transcriptional gene silencing[22,23] and manipulate jasmonic acid signaling[24–26] to subvert host defense.

DNA methylation can act as a defense mechanism against DNA viruses, including geminiviruses. Viral genomic DNA can be methylated[27–29], and therefore its replication is suppressed[30,31]. To counteract this inhibition, viruses encode suppressor proteins to interfere with DNA methylation[32–36] or the production of SAM[22,37–39]. It has been shown that βC1 can interact with and inactivate S-adenosyl homocysteine hydrolase (SAHH), an enzyme needed for SAM production, to suppress DNA methylation and enhance TYLCCNV virulence[22]. Thus far, whether active DNA demethylation plays a role in plant-virus interactions and whether the active DNA demethylation machinery is targeted by virus-encoded proteins remain unknown.

In this study, we investigated the role of active DNA demethylation in plant-virus interactions. We used TYLCCNV and its betasatellite TYLCCNB as a model system[20]. We found that active DNA demethylation promotes viral DNA hypomethylation and TYLCCNV/TYLCCNB virulence. βC1 interacts with NbROS1L in *Nicotiana benthamiana* and with DME in *Arabidopsis*. The interaction between βC1 and DME enhances the DNA glycosylase activity to counteract viral DNA methylation and increase viral virulence.

## Results

### Active DNA demethylation promotes viral DNA hypomethylation and TYLCCNV/TYLCCNB virulence in *N. benthamiana*.
By searching for ROS1 orthologs, we identified six DME-like (DML) DNA glycosylases in *N. benthamiana*, namely NbROS1, NbROS1L, NbDML3, NbDML4, NbDML5 and NbDML6 (Supplementary Fig. 1a). Except *NbDML6*, they appeared to be ubiquitously expressed in *N. benthamiana* plants (Supplementary Fig. 1b). We tested their role in the plant-virus interactions by knocking down their expression by using tobacco rattle virus (TRV)-induced gene silencing (VIGS) (Supplementary Figs. 2 and 3a). Plants infiltrated with pCambia1300 (TRV-free empty vector), with TRV-*GUS* (as a VIGS control) and with TRV-*NbDMLs* were mock-inoculated, inoculated with TYLCCNV alone, or inoculated with TYLCCNV plus TYLCCNB (TYLCCNV+B). pCambia1300-, TRV-*GUS*- or TRV-*NbDMLs*-infiltrated plants did not show any symptoms when inoculated with TYLCCNV alone (Fig. 1a and Supplementary Fig. 3b), consistent with the previous finding that TYLCCNV alone does not induce disease symptoms in *N. benthamiana*[20]. When inoculated with TYLCCNV+B, pCambia1300- and TRV-*GUS*-infiltrated plants showed dramatically curled leaves, whereas plants infiltrated with TRV-*NbDMLs* displayed only mild symptoms (Fig. 1a and Supplementary Fig. 3b). We next performed Southern blot and quantitative PCR (qRCR) analyses to determine the accumulation of viral DNA in infected leaves. Comparably low levels of TYLCCNV DNA accumulated in pCambia1300-, TRV-*GUS*- or TRV-*NbDMLs*-infiltrated plants when inoculated with TYLCCNV alone. High levels of TYLCCNV and TYLCCNB were accumulated in plants infiltrated with pCambia1300- or TRV-*GUS* then inoculated with TYLCCNV+B. However, the viral load was significantly decreased in plants infiltrated with TRV-*NbDMLs* (Fig. 1b, c and Supplementary Fig. 3c). These results validate that TYLCCNB promotes TYLCCNV infection[20], and suggest that active DNA demethylation plays a major role in the promotion of TYLCCNV infection by TYLCCNB.

Given that active DNA demethylation counteracts DNA methylation, we wondered whether removal of DNA methylation from viral genome is associated with the high accumulation of TYLCCNV/ TYLCCNB and increased symptom severity. To test this, we performed bisulfite sequencing to measure the methylation levels of TYLCCNV genomic DNA, represented by an intergenic region that contains the origin of replication and divergent viral promoters (Supplementary Fig. 3d). We found that ~23% C's in TYLCCNV were methylated in plants inoculated with TYLCCNV alone, regardless of whether *NbDMLs* were knocked down (Fig. 1d). The levels of methylation decreased to ~12% in plants infiltrated with pCambia1300 or TRV-*GUS* then inoculated with TYLCCNV+B. Knockdown of *NbDMLs* greatly restored methylation of the viral DNA (Fig. 1d). These results suggest that TYLCCNB causes a decrease in viral DNA methylation, which is mainly achieved through active DNA demethylation.

**βC1 interacts with NbROS1L.** βC1 can suppress DNA methylation through inactivating SAHH[22]. Given our finding that the TYLCCNB-mediated decrease in viral DNA methylation is mainly dependent on *NbDMLs*, we reasoned that βC1 could also target the active DNA demethylation machinery. The expression levels of *NbDMLs* were not significantly different in mock-, TYLCCNV- or TYLCCNV+B-inoculated plants (Supplementary Fig. 4a), suggesting that βC1 does not transcriptionally regulate *NbDMLs*. We then tested whether βC1 can interact with NbDMLs by performing bimolecular fluorescence complementation (BiFC) assays. We found that βC1 could interact with NbROS1L in the nucleus, but not with other members of the family (Fig. 2a and Supplementary Fig. 4b, c). Further analyses revealed that the N-terminal region of NbROS1L mediated its interaction with βC1 (Fig. 2b and Supplementary Fig. 4d). The interaction between βC1 and the N-terminal region of NbROS1L was confirmed by co-immunoprecipitation (Fig. 2c). We then

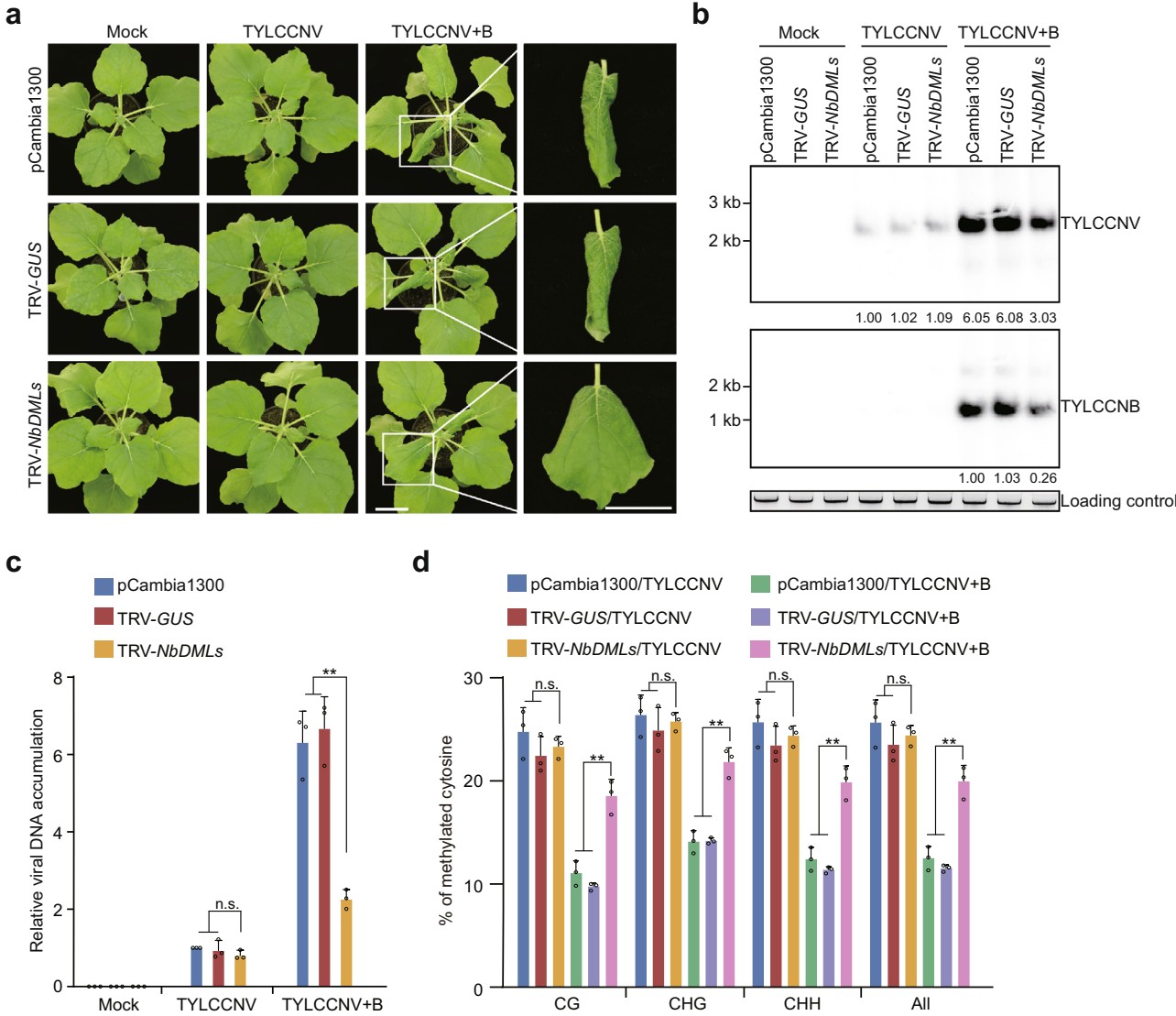

**Fig. 1 Active DNA demethylation promotes TYLCCNV+B virulence and viral DNA hypomethylation in *N. benthamiana*. a** Symptoms of pCambia1300-, TRV-*GUS*-, and TRV-*NbDMLs*-infiltrated *N. benthamiana* plants mock-inoculated, or inoculated with TYLCCNV or TYLCCNV+B. Photographs were taken at 10 days post inoculation (dpi). Scale bar, 2 cm. **b** Accumulation of TYLCCNV and TYLCCNB in the leaves of indicated plants at 10 dpi as determined by Southern blot. Total DNA was stained with ethidium bromide (EB) as loading control. **c** Accumulation of TYLCCNV in the leaves of indicated plants at 10 dpi as determined by qPCR. The coat protein gene of TYLCCNV was amplified. The *N. benthamiana 25S nuclear rRNA* gene (*Nb25SrRNA*) was used as an internal control. Values are means ± SD (*n* = 3 independent experiments). Statistical significance was determined using two-tailed Student's *t* test; \*\**p* < 0.01, ns not significant. **d** Percentage of methylated cytosines in the intergenic region (IR) of TYLCCNV in the leaves of indicated plants as determined by bisulfite sequencing. Values are means ± SD (*n* = 3 independent experiments). Statistical significance was determined using two-tailed Student's *t* test; \*\**p* < 0.01, ns not significant. Uncropped pictures, blots and gels for **a** and **b**, raw data and *p* values for **c** and **d** are provided in the source data.

mapped the region of βC1 that is required for its interaction with NbROS1L, and identified a 10-amino acid motif, from residue 11 to 20, essential for the interaction (Supplementary Fig. 5a, b). Further mutational analysis showed that replacement of the Val in position 17 of βC1 with Ala (V17A) abolished the interaction with NbROS1L (Fig. 2d, e and Supplementary Fig. 5c), but did not affect its interaction with SAHH (Supplementary Fig. 5d, e), indicating that V17 is specifically required for the interaction between βC1 and NbROS1L (Fig. 2d, e).

**The βC1-NbROS1L interaction is important for TYLCCNV+B virulence and viral DNA hypomethylation.** We next investigated whether the βC1-NbROS1L interaction is important for the TYLCCNB-mediated methylation decrease and promotion of

virulence. For this purpose, we generated a TYLCCNB clone expressing βC1$^{V17A}$. Interestingly, plants inoculated with TYLCCNV+B$^{V17A}$ showed very mild symptoms and lower levels of TYLCCNV/TYLCCNB accumulation (Fig. 3a–c and Supplementary Fig. 6a, b); concomitantly, higher levels of viral DNA methylation were detected (Fig. 3d). These results suggest that the interaction between βC1 and NbROS1L is important for the TYLCCNB-mediated decrease in viral DNA methylation and TYLCCNV/TYLCCNB virulence.

**The βC1-DME interaction promotes BSCTV virulence and viral DNA hypomethylation in *Arabidopsis*.** To examine whether the role of βC1 in regulating active DNA demethylation and viral infection is conserved in other plant species, we tested

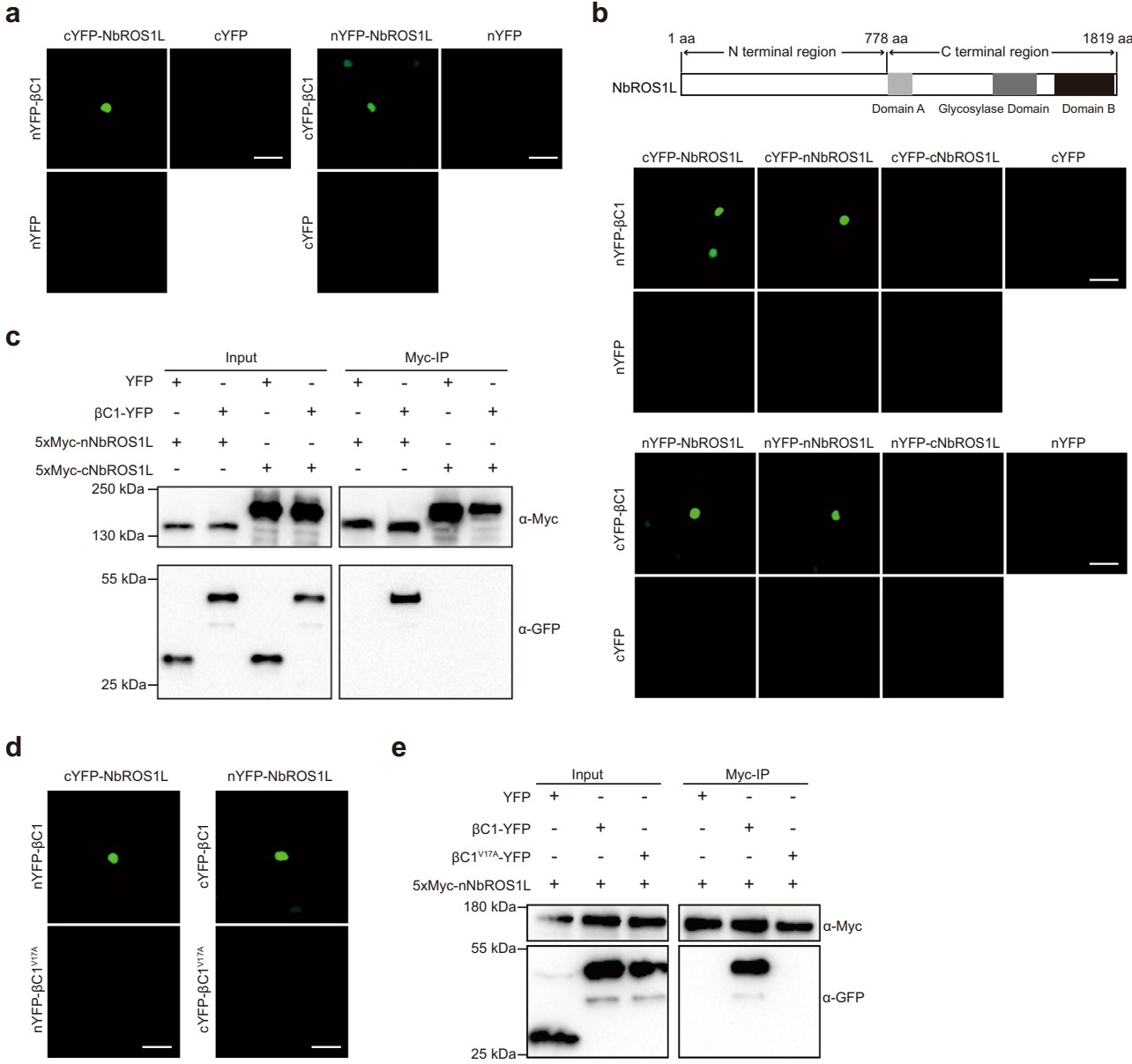

**Fig. 2 βC1 interacts with NbROS1L. a** BiFC analysis of the interaction of βC1 with NbROS1L. Scale bars, 50 μm. **b** BiFC analysis of the interactions of βC1 with the N-terminal region of NbROS1L (nNbROS1L) and the C-terminal region of NbROS1L (cNbROS1L). Scale bars, 50 μm. Schematic representation of NbROS1L protein structure is shown above. **c** Co-immunoprecipitation analysis of the interactions of βC1 with nNbROS1L and cNbROS1L. YFP was used as a negative control. **d** BiFC analysis of the interaction between βC1$^{V17A}$ and NbROS1L. Scale bars, 50 μm. **e** Co-immunoprecipitation analysis of the interaction between βC1$^{V17A}$ and nNbROS1L. YFP was used as a negative control. Images in **a**–**e** are representative of three independent experiments. Raw data for **a**, **b** and **d** and uncropped blots for **c** and **e** are provided in the source data.

if βC1 can interact with *Arabidopsis* DNA glycosylases. BiFC results revealed that βC1 could interact with DME (Supplementary Fig. 7a). Similar to what was observed for NbROS1L, we found that the N-terminal region of DME and the Val in position 17 of βC1 were required for the interaction (Fig. 4a and Supplementary Fig. 7b–e). We next determined whether the βC1-DME interaction promotes viral virulence. With this aim, we generated transgenic *Arabidopsis* expressing βC1 or βC1$^{V17A}$, and then inoculated these plants with the *Arabidopsis*-infecting geminivirus beet severe curly top virus (BSCTV) (Supplementary Fig. 8a). Wild-type (Col-0) plants developed mild curly top symptoms after BSCTV infection, while the βC1-expressing plants developed dramatically more

severe symptoms (Fig. 4b and Supplementary Fig. 8b). These symptoms were largely ameliorated when βC1$^{V17A}$ was expressed instead (Fig. 4b and Supplementary Fig. 8b). Consistent with this, a high level of BSCTV DNA was accumulated in the βC1-expressing plants, while viral accumulation was decreased in the βC1$^{V17A}$-expressing plants (Fig. 4c, d and Supplementary Fig. 8c). The high accumulation of BSCTV DNA and the development of severe symptoms in βC1-expressing plants were associated with hypomethylation of BSCTV DNA, which was restored to a great extent in βC1$^{V17A}$-expressing plants (Fig. 4e). These results suggest that the βC1-DME interaction is important for BSCTV DNA hypomethylation and virulence in *Arabidopsis*.

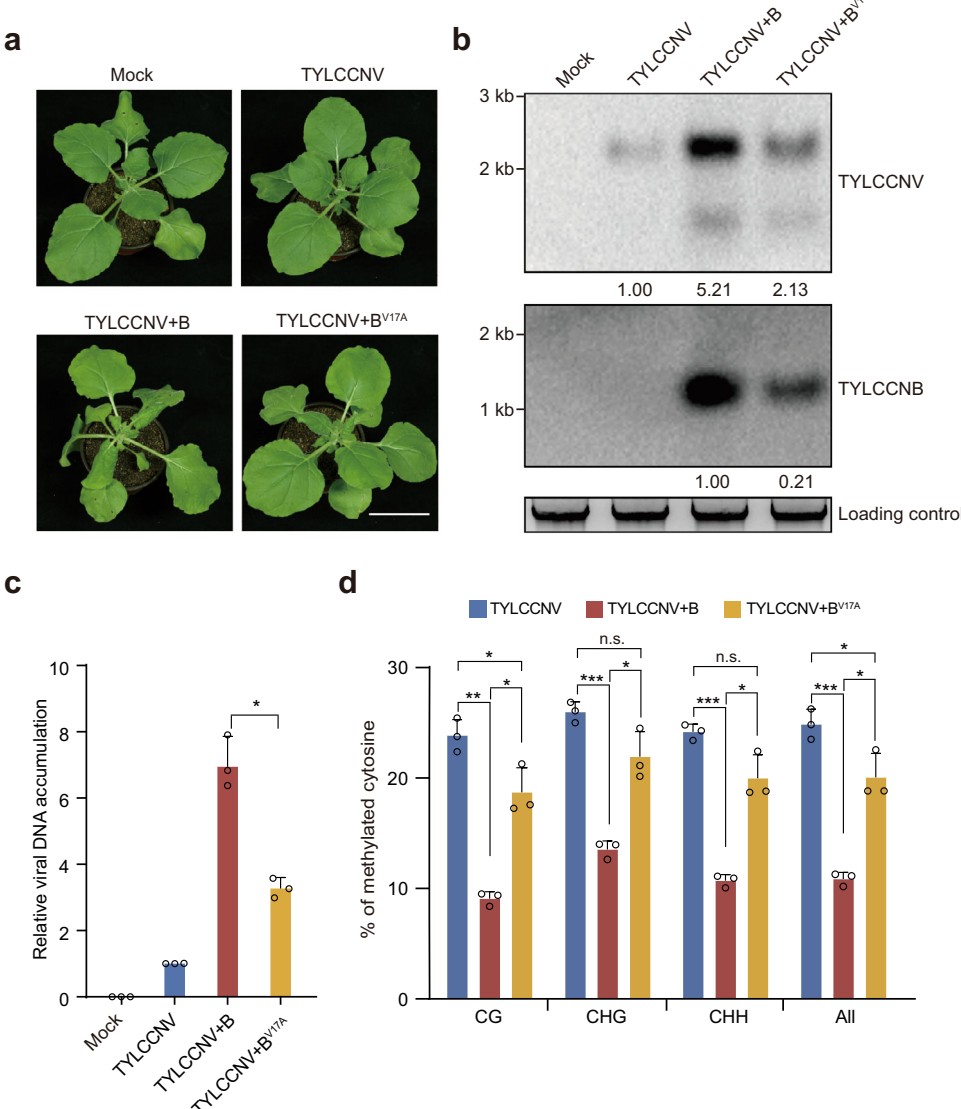

**Fig. 3 The βC1-NbROS1L interaction is important for TYLCCNV+B virulence and viral DNA hypomethylation. a** Symptoms of *N. benthamiana* plants mock-inoculated or inoculated with TYLCCNV, TYLCCNV+B or TYLCCNV+B$^{V17A}$ at 10 dpi. Scale bar, 2 cm. **b** Accumulation of TYLCCNV and TYLCCNB in the leaves of indicated plants at 10 dpi as determined by Southern blot. Total DNA was stained with EB as loading control. **c** Accumulation of TYLCCNV in the leaves of indicated plants at 10 dpi as determined by qPCR. The coat protein gene of TYLCCNV was amplified. The *N. benthamiana 25S* nuclear *rRNA* gene (*Nb25SrRNA*) was used as an internal control. Values are means ± SD (*n* = 3 independent experiments). Statistical significance was determined using two-tailed Student's *t* test; *$p < 0.05$. **d** Percentage of methylated cytosines in the intergenic region (IR) of TYLCCNV in the leaves of indicated plants at 10 dpi as determined by bisulfite sequencing. Values are means ± SD (*n* = 3 independent experiments). Statistical significance was determined using two-tailed Student's *t* test; *$p < 0.05$, **$p < 0.01$, ***$p < 0.001$, ns not significant. Uncropped pictures, blots and gels for **a** and **b**, raw data and *p* values for **c** and **d** are provided in the source data.

**βC1 promotes DME activity in vitro and in vivo.** DME has both DNA glycosylase and apyrimidinic lyase activities, which mediate the removal of 5mC from double-stranded DNA and the cleavage of the DNA backbone at the abasic site to produce β- and β, δ-elimination products, respectively[40,41]. Because the βC1-DME interaction promotes viral DNA hypomethylation, we wondered whether the interaction can directly enhance the activity of DME. To test this, recombinant DME, GST-βC1, and GST-βC1$^{V17A}$ were expressed and purified (Supplementary Fig. 9a). A 5′-fluorescein-labeled oligonucleotide substrate containing 5mC was incubated with DME plus varying amounts of GST, GST-βC1, or GST-βC1$^{V17A}$. β- and β, δ-elimination products were yielded when the substrate was

incubated with DME. The amount of products was increased when GST-βC1 was added; however, this was abolished when GST-βC1 was replaced with GST-βC1$^{V17A}$ (Fig. 5a, b and Supplementary Fig. 9b). These results suggest that βC1 can directly enhance the activity of DME through protein–protein interaction.

Given that DME activity was enhanced by βC1 in vitro, we asked whether this would also occur in vivo. To test this possibility, we generated transgenic *N. benthamiana* plants overexpressing *DME* (Supplementary Fig. 10a) and inoculated these plants with TYLCCNV alone, TYLCCNV+B, or TYLCCNV+B$^{V17A}$. We found that DME-overexpressing plants, like wild-type plants, did not show any symptoms when inoculated with TYLCCNV alone

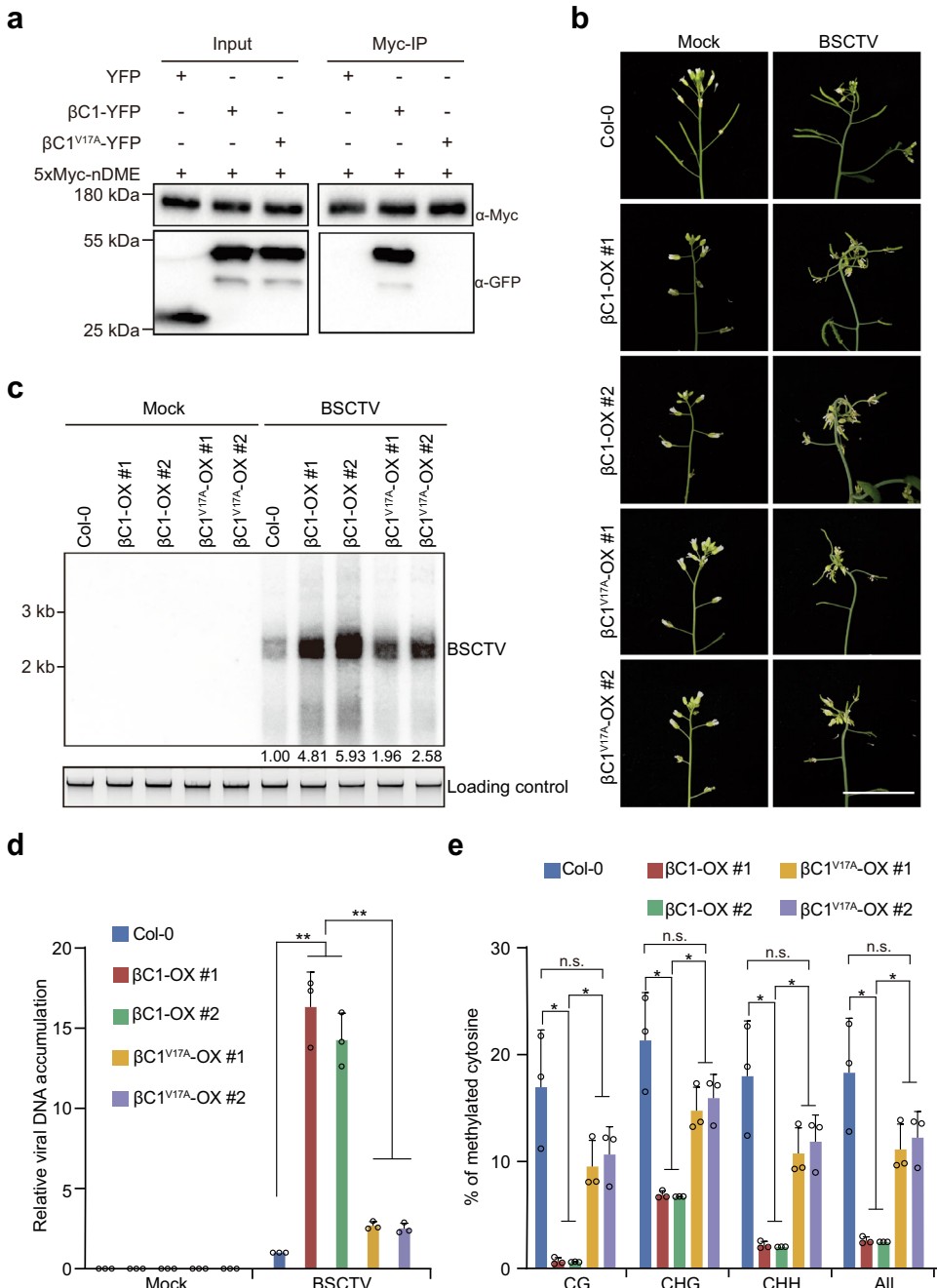

**Fig. 4 The βC1-DME interaction promotes BSCTV virulence in *Arabidopsis*. a** Co-immunoprecipitation analysis of the interactions of βC1 or βC1[V17A] with the N-terminal region of DME (nDME). YFP was used as a negative control. Results are representative of three independent experiments. **b** Symptoms of Col-0, βC1-expressing lines and βC1[V17A]-expressing lines inoculated with BSCTV at 10 dpi. Scale bars, 2 cm. **c** Accumulation of BSCTV in the indicated plants at 10 dpi as determined by Southern blot. Total DNA was stained with EB as loading control. **d** Accumulation of BSCTV in the indicated plants at 10 dpi as determined by qPCR. The coat protein gene of BSCTV was amplified. The *Arabidopsis 25S nuclear rRNA* gene (*At25SrRNA*) was used as an internal control. Values are means ± SD ($n = 3$ independent experiments). Statistical significance was determined using two-tailed Student's $t$ test; $**p < 0.01$. **e** Percentage of methylated cytosines in the intergenic region (IR) of BSCTV in the indicated plants at 10 dpi as determined by bisulfite sequencing. Values are means ± SD ($n = 3$ independent experiments). Statistical significance was determined using two-tailed Student's $t$ test; $*p < 0.05$, ns not significant. Uncropped pictures, blots and gels for **a**, **b** and **c**, raw data and $p$ values for **d** and **e** are provided in the source data.

(Fig. 5c and Supplementary Fig. 10b). However, they developed more severe symptoms than wild-type plants when inoculated with TYLCCNV+B, an effect that was abolished when the plants were inoculated with TYLCCNV+B[V17A] (Fig. 5c and Supplementary Fig. 10b). Consistent with this, *DME*-overexpressing plants inoculated with TYLCCNV+B had lower levels of viral DNA

methylation and accumulated higher levels of TYLCCNV/B, compared to those inoculated with TYLCCNV+B[V17A] (Fig. 5d–f and Supplementary Fig. 10c). These data suggest that overexpression of DME enhances TYLCCNV infection by further decreasing the level of viral DNA methylation, and that the interaction between βC1 and DME is required for this process.

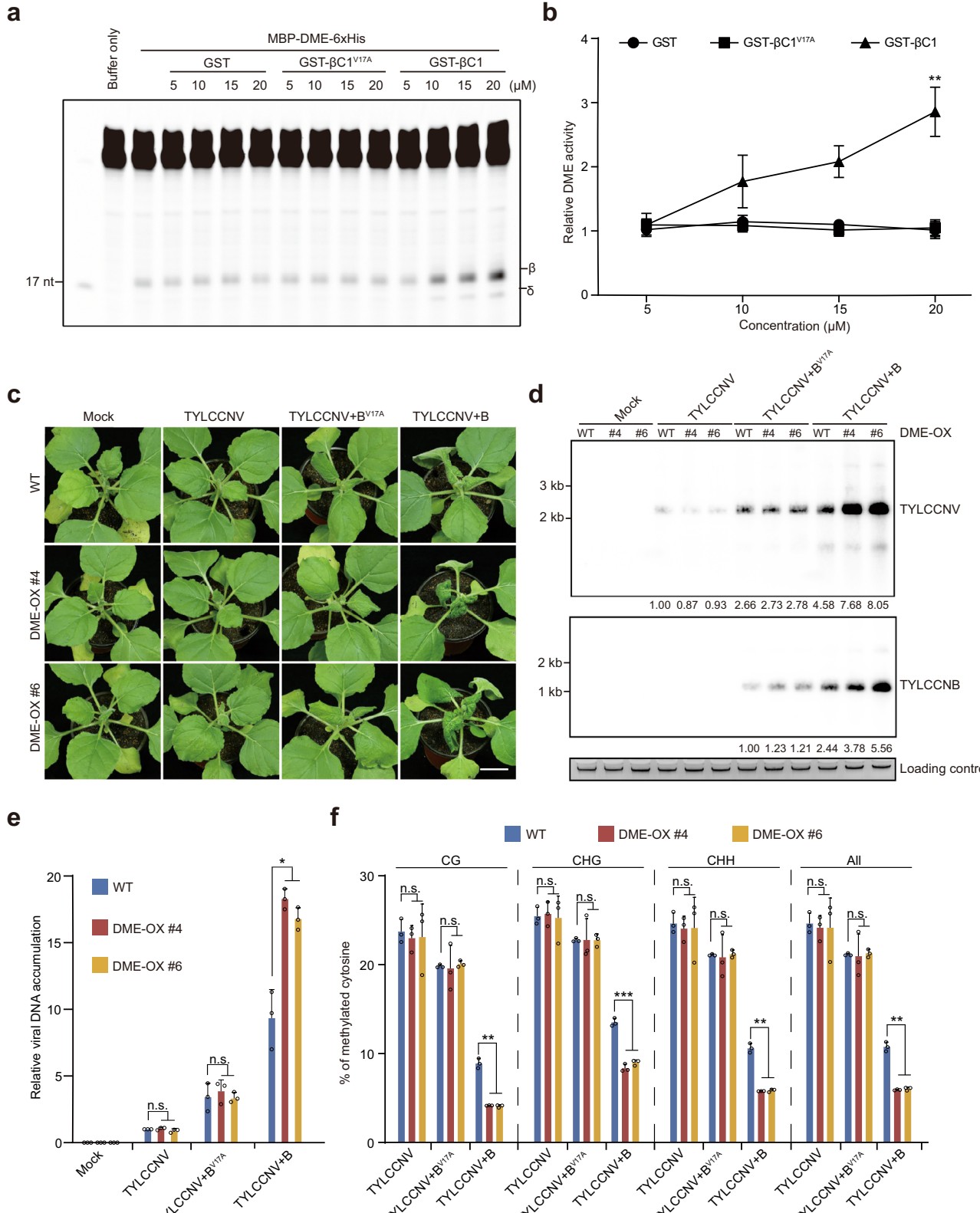

## Discussion

Active DNA demethylation plays an important role in shaping genomic DNA methylation patterns and regulating transcription of many endogenous loci[1]. In this study, we show that active DNA demethylation targets DNA of TYLCCNV and its betasatellite TYLCCNB to promote their virulence. βC1 regulates the activity or targeting of active DNA demethylation for

suppression of viral DNA methylation and enhancement of viral virulence.

*DME* was originally identified by mutations that cause maternal effects on seed viability[11], and has been thought to be expressed and function specifically in reproductive tissues during gametogenesis for gene and transposon activation[11,13,17]. However, *DME* has been detected to be constitutively expressed

**Fig. 5 βC1 promotes DME activity in vitro and in vivo. a** In vitro 5mC excision activity of DME protein. Double-stranded oligonucleotide substrate (one strand methylated) was incubated with DME and varying amounts of GST, GST-βC1 or GST-βC1$^{V17A}$. 17-nucleotide size marker and β- and δ-elimination products are indicated. **b** Quantification of relative DME activity. Values are means ± SD ($n = 3$ independent experiments). Statistical significance was determined using two-tailed Student's $t$ test; $**p < 0.01$. **c** Symptoms of wild-type and DME-overexpressing *N. benthamiana* lines mock-inoculated or inoculated with TYLCCNV, TYLCCNV+B or TYLCCNV+B$^{V17A}$ at 10 dpi. Scale bar, 2 cm. **d** Accumulation of TYLCCNV and TYLCCNB in the leaves of indicated plants at 10 dpi as determined by Southern blot. Total DNA was stained with EB as loading control. **e** Accumulation of TYLCCNV in the leaves of indicated plants at 10 dpi as determined by qPCR. The coat protein gene of TYLCCNV was amplified. The *N. benthamiana 25S nuclear rRNA* gene (*Nb25SrRNA*) was used as an internal control. Values are means ± SD ($n = 3$ independent experiments). Statistical significance was determined using two-tailed Student's $t$ test; $*p < 0.05$, ns not significant. **f** Percentage of methylated cytosines in the intergenic region (IR) of TYLCCNV in the leaves of indicated plants at 10 dpi as determined by bisulfite sequencing. Values are means ± SD ($n = 3$ independent experiments). Statistical significance was determined using two-tailed Student's $t$ test; $**p < 0.01$, $***p < 0.001$, ns not significant. Uncropped pictures, blots and gels for **a**, **c** and **d**, raw data and $p$ values for **b**, **e** and **f** are provided in the source data.

in vegetative tissues by microarray, RNA-sequencing, RT-PCR and GUS reporter analyses[12,42–45]. *DME* also functions in vegetative tissues. Knockdown of *DME* expression leads to DNA hypermethylation at hundreds of genomic regions, which eventually leads to downregulated expression of dozens of genes under normal growth conditions[45] and compromised activation of defense-related genes and increased plant susceptibility to infection when the plants are invaded by bacterial or fungal pathogens[44,45]. Our data suggest that DME can be activated by a viral suppressor protein to reduce DNA methylation level of the viruses in plant leaves. Our findings provide evidence that DME expression and function extend beyond reproductive tissues in *Arabidopsis*. In *M. truncatula*, DME is expressed in the root nodules, an organ formed in legumes through symbiotic interaction with nitrogen-fixing rhizobia, and DME expression is required for activation of many nodule differentiation-associated genes and normal development of nodules[46]. It will be interesting to investigate whether DME also targets genes of rhizobia and how its activity is regulated.

We showed that TYLCCNB is required for active DNA demethylation activity on viral DNA. We further found that βC1 interacts with DME and enhances the DNA demethylation activity of DME in vitro (Figs. 4a and 5a, b), suggesting that βC1 may increase NbROS1L and DME activity on viral DNA. This could be achieved through inducing structural or conformational changes. However, we cannot at this point exclude the possibility that βC1 may facilitate NbROS1L or DME targeting to the viral genome. Supporting this possibility, we found that loss of *NbROS1L* or overexpression of *DME* has no effect on viral DNA methylation level, viral replication, and virulence, in the absence of TYLCCNB or βC1 (Figs. 1 and 5). It is possible that access of NbROS1L/DME to the viral DNA requires assistance from another protein. Since βC1 has been shown to bind both single-stranded and double-stranded DNA in vitro, albeit lacking size or sequence specificity[47], it could provide such function. Furthermore, βC1 can substitute for the DNA B component of tomato leaf curl New Delhi virus (ToLCNDV), a bipartite begomovirus, to help the DNA A component of ToLCNDV move and sustain systemic infection of tomato plants[48], indicating that βC1 likely has DNA-binding ability in vivo. Nonetheless, it is possible that active DNA demethylation could also target plant endogenous genes to promote viral virulence.

We have previously found that βC1 also interacts with and inactivates SAHH to reduce viral DNA methylation[22]. The V17A mutation of βC1 disrupts the interaction between βC1 and NbROS1L or DME but does not affect the interaction between βC1 and SAHH, allowing us to uncouple the effects caused by βC1-mediated activation of NbROS1L or DME and those caused by βC1-mediated suppression of SAHH. We found that TYLCCNV DNA methylation level is lower when βC1 carries the V17A mutation (Fig. 3d) or when *NbDMLs* are silenced compared to that in TYLCCNV-inoculated control (Fig. 1d).

Similarly, BSCTV DNA methylation level is lower in βC1$^{V17A}$-OX plants than in Col-0 (Fig. 4e). These results suggest that βC1-mediated suppression of SAHH activity also contributes to βC1-mediated suppression of DNA methylation. Thus, DNA viruses have evolved to both suppress DNA methylation and boost active DNA demethylation as counter-defense strategies.

## Methods

**Plant materials and growth conditions.** The *35S::RFP-H2B* transgenic *N. benthamiana* line was previously described[49]. Other plant materials, including *35S::5×Myc-DME N. benthamiana* transgenic line and *35S::3×Flag-βC1* and *35S::3×Flag-βC1$^{V17A}$ Arabidopsis* transgenic lines, were generated in this study. *N. benthamiana* plants were grown in a controlled growth chamber at 25 °C under a 16 h light/8 h dark photoperiod and *Arabidopsis* plants were grown in a growth room under long photoperiod conditions (16 h light, 22 °C/8 h dark, 18 °C).

**Cloning.** cDNAs of *NbROS1*, *NbROS1L*, *NbDML3*, *NbDML4*, *NbDML5* and *NbDML6* were PCR-amplified, cloned into the pLB vector (TIANGEN, VT205) and sequenced. Three cDNA fragments of *NbROS1*, *NbDML3*, and *NbDML4* were fused by overlap PCR and the fusion product was cloned into the pTRV2 vector[50], generating the TRV-based VIGS vector pTRV2-*NbDMLs* for knockdown of *NbDMLs*. For BiFC, the coding sequence of βC1 and cDNAs of *NbROS1*, *NbROS1L*, *NbDML3*, *NbDML4*, *NbDML5*, *NbDML6*, *ROS1*, *DML2*, *DML3* and *DME* were PCR-amplified and cloned into p2YN and p2YC. For mapping the regions of NbROS1L and DME that mediate the interactions with βC1, cDNAs corresponding to the N-terminal and C-terminal regions of NbROS1L and DME were PCR-amplified and cloned into p2YN and p2YC. For mapping the region and residue of βC1 that mediate the interactions with NbROS1L and DME, the coding sequence of βC1 was PCR-amplified and cloned into the pLB vector. The construct was used to generate all the truncated and mutant forms of βC1 using the KOD-Plus-Mutagenesis Kit (TOYOBO, SMK-101). For co-immunoprecipitation, the coding sequences of βC1 and βC1$^{V17A}$ were PCR-amplified and cloned into pCambia1300-35S-YFP and pCambia1300-35S-3×Flag to generate 35S-βC1-YFP, 35S-βC1$^{V17A}$-YFP, 35S-3×Flag-βC1, and 35S-3×Flag-βC1$^{V17A}$ constructs. cDNAs corresponding to the N-terminal and C-terminal regions of NbROS1L were PCR-amplified and cloned into pCambia1300-35S-5×Myc, generating 35S-5×Myc-nNbROS1L and 35S-5×Myc-cNbROS1L constructs. cDNAs corresponding to *DME* and the N-terminal regions of *DME* were PCR-amplified and cloned into pCambia1300-35S-5×Myc, generating 35S-5×Myc-DME and 35S-5×Myc-nDME constructs. For the evaluation of TYLCCNV+B$^{V17A}$ virulence, the sequence of TYLCCNB was cloned into the pLB vector and the construct was used to introduce the βC1$^{V17A}$ point mutation, generating pLB-TYLCCNB$^{V17A}$. The sequence of TYLCCNB$^{V17A}$ was PCR-amplified and cloned into pBinPLUS to generate pBinPLUS-TYLCCNB$^{V17A}$ using the ClonExpress II One Step Cloning kit (Vazyme, C112-02). pLB-TYLCCNB$^{V17A}$ was digested with *Kpn*I and *Eco*RI and the product was cloned into pBinPLUS-TYLCCNB$^{V17A}$, generating pBinPLUS-2TYLCCNB$^{V17A}$. For the expression of recombinant proteins in *E. coli*, the coding sequences of βC1 and βC1$^{V17A}$ were PCR-amplified and cloned into the pGEX4T-3 vector and the coding sequence of *DME* were PCR-amplified and cloned into the pET11-6×His-MBP vector[51].

**Agroinfiltration and virus inoculation.** All binary plasmids were individually transformed into *Agrobacterium tumefaciens* strain EHA105. The transformants were cultured in liquid LB with appropriate antibiotics, collected by centrifugation and resuspended in the infiltration buffer (10 mM MgCl$_2$, 10 mM MES pH 5.6, 100 mM acetosyringone). The suspensions were placed in dark for 2 h at room temperature before infiltration.

For TYLCCNV inoculation following VIGS, *A. tumefaciens* cultures carrying pCambia1300, pTRV1 plus pTRV2-*GUS* (TRV-*GUS*) or pTRV1 plus pTRV2-*NbDMLs* (TRV-*NbDMLs*) at an OD600 = 0.2 were inoculated into *N. benthamiana*

plants at the eight-leaf stage. One week later, *A. tumefaciens* cultures harboring pBinPLUS (mock) or infectious clones of TYLCCNV or TYLCCNV plus TYLCCNB (TYLCCNV+B) at an OD600 = 1.0 were inoculated into pCambia1300-, TRV-*GUS*- or TRV-*NbDMLs*-infiltrated plants. For other TYLCCNV inoculation, *N. benthamiana* plants at the eight-leaf stage were directly agroinoculated with infectious clones of TYLCCNV, TYLCCNV+B or TYLCCNV +B^V17A. For BSCTV inoculation, 4-week-old *Arabidopsis* plants were inoculated during bolting period and the bolts were cut at their base. The cut was applied with *A. tumefaciens* cultures carrying pCambia1300 (mock) or infectious clones of BSCTV and then insect pins were used to puncture the cut multiple times[27]. Symmetrically infected leaves were harvested at 10 days post inoculation.

**RNA extraction and RT-qPCR**. RNA was isolated using an Eastep Super Total RNA Extraction kit (Promega, LS1040) and 1 μg of total RNA was reversely transcribed by M-MLV reverse transcriptase (Invitrogen, 28025013). qPCR was conducted using SYBR Premix EX Taq (TaKaRa, RR420A). *NbACT2* was used an internal control. Primer sequences are listed in Supplementary Table 1.

**DNA extraction, Southern blot analysis, and qPCR**. Total DNA was extracted from infected plant leaves using Plant Genomic DNA Kit (TIANGEN, DP305). For detection of viral DNA accumulation by Southern blot, 1 μg of total DNA was loaded onto 1.0% agarose gels and separated by electrophoresis. Agarose gels were denatured with denaturing buffer (0.5 M NaOH, 1.5 M NaCl) for 40 min and neutralized twice with neutralizing buffer (0.5 M Tris-HCl pH 7.0, 1.5 M NaCl) for 20 min. After denaturation and neutralization, total DNA was transferred to Hybond N+ nylon membranes (GE Healthcare, RPN303B). TYLCCNV+B and BSCTV DNA probes were labeled with [α-$^{32}$P] dCTP using Random Primer DNA Labeling Kit Ver. 2 (TaKaRa, 6045) or with digoxigenin (DIG) using DIG High Prime DNA labeling and detection starter kit II (Roche, 11585614910). Membranes were hybridized at 65 °C overnight and washed three times with 1 × SSC, 0.1% SDS at 65 °C for 15 min. The radioactive signal was detected by Personal Molecular Imager (Bio-Rad) and the signal obtained with the DIG-labeled probes was detected by Image Lab (Bio-Rad). For detection of viral DNA accumulation by qPCR, the coat protein genes of TYLCCNV and BSCTV were amplified as described previously[52]. Primer sequences for generating the probes are listed in Supplementary Table 1.

**DNA bisulfite sequencing analysis**. Total DNA (500 ng) was subjected to bisulfite treatment using EZ DNA Methylation Gold^TM Kit (Zymo Research, D5005), according to the manufacturer's instructions. The bisulfite-treated DNA was used as templates for amplification of the IR region of TYLCCNV or BSCTV. Primer sequences are listed in Supplementary Table 1. The products were purified with AMPure XP beads (Beckman, A63880) and the purified DNA products were PCR-amplified and the adapters were added using GXL DNA polymerase (Takara, R050Q) and primer F (5′-AATGATACGGCGACCACCGAGATCTACAC-Index-TCGTCGGCAGCGTC-3′) and primer R (5′-CAAGCAGAAGACGGCATACGA GAT-Index-GTCTCGTGGGCTCGG)[53]. The final products were purified with AMPure XP beads for sequencing on an Illumina HiSeqX-Ten platform by Annoroad Gene Technology (Beijing). PCR-duplicated reads were removed and the unique reads with read counts above 5 were retained for cytosine methylation analysis using Kismeth (http://katahdin.mssm.edu/kismeth/revpage.pl).

**BiFC assay**. Equal volumes of *A. tumefaciens* cultures at an OD600 = 1.0. were mixed prior to infiltration. Leaves of 4-week-old *35S::RFP-H2B* transgenic *N. benthamiana* line were infiltrated with *A. tumefaciens* cultures carrying pairs of BiFC constructs. Epidermal cells of the assayed leaves were examined by confocal laser scanning microscopy 980 (Zeiss, GER) 48 h post infiltration. For confocal imaging, the excitation wavelength for YFP was set at 514 nm and the emission was captured at 565–585 nm, and the excitation wavelength for RFP was set at 543 nm and the emission was captured at 590–630 nm. *RFP-H2B* signals were used to mark the nucleus.

**Co-immunoprecipitation**. Four-week-old *N. benthamiana* leaves were infiltrated with *A. tumefaciens* cultures carrying the pCambia1300 constructs. Forty-eight hours post infiltration, the leaves were collected and ground in liquid nitrogen. Total proteins were extracted from 2 g of ground powder in 4 ml of IP buffer (50 mM Tris-HCl pH 7.5, 150 mM NaCl, 5 mM MgCl$_2$, 2 mM EDTA, 0.5% Triton X-100, 5% glycerol, 5 mM DTT, 1 mM PMSF, and 1 Protease Inhibitor Cocktail tablet (Roche, 4693116001)). The extract was centrifuged three times at 20,000 × *g* for 10 min each time at 4 °C and the supernatant was collected. Twenty-five microliters of anti-c-myc-agarose affinity gel (Sigma–Aldrich, A7470) or GBP beads (homemade) were added to the supernatant and incubated at 4 °C for 2 h with rotation. After incubation, the beads were washed three times with IP buffer at 4 °C. Finally, 50 μl of 2 × SDS loading buffer were added to the sample for western blot analyses.

**Western blot analysis**. Proteins were separated by SDS-PAGE and transferred to nitrocellulose membranes (GE Healthcare, 10600003). Membranes were incubated with primary antibodies and secondary antibodies sequentially. Primary antibodies we used include anti-GFP (Roche, 11814460001, 1:5000 dilution), anti-c-myc

(Roche, 11667203001, 1:2000 dilution), anti-Flag M2 (Sigma, F1804, 1:5000 dilution), anti-HA (Roche, 11666606001, 1:5000 dilution), anti-actin (Sangon, D191048, 1:5000 dilution) and anti-α-tubulin (Sigma, T5168, 1:5000 dilution). Horseradish peroxidase-conjugated goat anti-mouse IgG (Sigma, A4416, 1:5000 dilution) was used as the secondary antibody. ECL prime western blot detection reagent (GE Healthcare, RPN2232) was added to the membranes for chemiluminescence detection using Image Lab (Bio-Rad).

**Expression and purification of recombinant proteins**. The constructs expressing βC1, βC1^V17A and DME were transformed into *E. coli* BL21 (DE3) cells and transformants were grown at 37 °C until the OD$_{600}$ reached 0.6. Protein expression was induced by the addition of 0.5 mM isopropyl β-D-1-thiogalactopyranoside at 18 °C for 16 h. Cells were harvested and resuspended in lysis buffer (40 mM Tris-HCl pH 8.0, 500 mM NaCl, 10% glycerol and 1 mM PMSF). The suspension was lysed by sonication and centrifuged at 20,000 g for 1 h at 4 °C. For purification of His-tagged proteins, the supernatant was loaded onto a column packed with Ni-NTA. After washing with washing buffer (40 mM Tris-HCl pH 8.0, 500 mM NaCl, 20 mM imidazole), proteins were eluted with elution buffer (40 mM Tris-HCl pH 8.0, 500 mM NaCl, and 500 mM imidazole). For purification of GST-tagged proteins, the supernatant was loaded onto a column packed with glutathione sepharose 4B agarose beads (GE Healthcare, 17-0756-01). The beads were washed with washing buffer (40 mM Tris-HCl pH 8.0, 500 mM NaCl) and proteins were eluted with elution buffer (40 mM Tris-HCl pH 8.0, 500 mM NaCl and 20 mM glutathione reduced). For purification of both His-tagged and GST-tagged proteins, gel filtration was performed on Superdex 200 increase size exclusion columns (GE Healthcare, 28-9909-44) with gel filtration buffer (40 mM Tris-HCl pH 8.0, 150 mM NaCl and 1 mM DTT) and peak fractions were collected.

**DME activity assay**. DME activity assay was performed as described[54]. A pair of Cy5 end-labeled oligonucleotides (one strand methylated) were synthesized and annealed, generating double-stranded oligonucleotide substrate. The substrate was incubated with 0.5 μM DME plus varying amounts of GST, GST-βC1 or GST-βC1^V17A in a 15-μl reaction containing 40 mM Tris-HCl pH 8.0, 0.1 M KCl, 0.1 mM EDTA, 0.5 mM DTT, and 200 μg/ml BSA at 37 °C for 30 min. The reaction was terminated by the addition of 15 μl of loading buffer (95% formamide, 0.025% bromophenol blue, 0.025% SDS, 18 mM EDTA). The sample was heated at 95 °C for 10 min, and then chilled on ice. The sample was loaded onto 15% denaturing polyacrylamide gels and reaction products were separated by electrophoresis. Cy5 signal was detected using Typhoon (GE Healthcare) and quantified using ImageJ.

**Reporting summary**. Further information on research design is available in the Nature Research Reporting Summary linked to this article.

## Data availability

Bisulfite sequencing dataset generated in this study can be found in the NCBI Gene Expression Omnibus under accession number GSE188968. A reporting summary for this article is available as a Supplementary Information file. Source data are provided with this paper.

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

## Acknowledgements

This work is funded by grants from National Natural Science Foundation of China (31720103914, 31930089 and 31788103).

## Author contributions

X.Z. and Y.Q. designed the experiments; X.G. performed the research; X.G., C.L., Y.Q., and X.Z. analyzed the data and wrote the paper.

## Competing interests

The authors declare no competing interests.
