## [Peer Review File · Nature Communications]

Geminiviruses employ host DNA glycosylases to subvert DNA methylation-mediated defenseREVIEWER COMMENTS

Reviewer #1 (Remarks to the Author):

In this paper, Xiaojian Gui and colleagues provide compelling evidence that active viral demethylation is critical to promote virulence in coinfections by geminiviruses and satellite DNA. They show that the β C1 protein of the satellite TYLCCNB interacts with ROS1L and DEMETER (DME) glycosylases in *Nicotiana benthamiana* and *Arabidopsis thaliana*, respectively, to decrease viral DNA methylation and promote viral accumulation and virulence. This is a very nice, elegant and convincing paper. Very well done and very well written. I don't see any flaws and the data is of great significance in the field. I only have a few minor comments to do.

1) L76. In these experiments we can see the effect of TYLCCNB on TYLCCNV-infected plants that have been previously infiltrated with TRV-GUS or TRV-NbDMLs. I miss however a TRV-free control for comparison (as shown in Fig. 3). This control could be important as it has been shown that TRV interferes with DNA de/methylation (Diezma-Navas et al. *Mol Plant Pathol.* 2019 Oct;20(10):1439-1452. doi: 10.1111/mpp.12850. Epub 2019 Jul 5. PMID: 31274236). Is there any possible scenario by which TRV itself could influence the outcome of subsequent infections in this experiment?

2) L124. Without a proper TYLCCNV-infected control (this control is lacking) they cannot discard a transcriptional activation of NbDML in TYLCCNV-infected plants, and that NbDML activation be further modulated when TYLCCNB is co-infecting.

3) Interestingly, viral accumulation in TYLCCNV+B-infected plants remains much higher than controls (TYLCCNV-infected plants) when β C1 carries the V17A mutation (Fig. 3C) or when DMLs are silenced (Fig. 1C). It seems likely that ROS1L and β C1 partially retain their functions in both cases. However, methylation levels are comparable to that observed in TYLCCNV-infected controls. Is it possible that additional mechanisms (perhaps involving other DMLs) may assist β C1 in promoting virulence?

4) Viral DNA methylation in the presence of β C1(V17A) is similar to that observed in the TYLCCNV control. This mutation does not compromise the interaction between β C1 and SAHH. This seems to suggest that β C1-mediated suppression of DNA methylation via SAHH interaction is negligible (only CHH methylation is slightly reduced). However, is it possible that suppression of DNA methylation by SAHH and active demethylation by ROS1L/DME are coupled in a way that the reduced methylation observed TYLCCNV/BSCTV+B-infected plants is a consequence of both activities?

5) Please, the 35S::RFP-H2B transgenic *N. benthamiana* plants are not mentioned in the main text. Please, describe briefly and indicate what they were used for. In general, details provided in the methods section are insufficient to reproduce the experiments.

Reviewer #2 (Remarks to the Author):

This is review of the manuscript “Geminiviruses hijack host DNA glycosylases to subvert DNA methylation-mediated defense” submitted by Gui and colleagues to Nature Communications. In this study, the authors report that active DNA demethylation can target viral DNAs to promote their virulence. They also found that β C protein could directly interact with NbROS1L and DME from *Arabidopsis*, and enhanced the DNA glycosylase activity in vitro. In addition, they demonstrated that the protein interaction between β C and DME was essential for promote viral virulence. This study suggested a novel function of active DNA demethylation in plants. However, the evidence to support their hypothesis is not enough so far.

Specific comments:

1. The DNA methylation changes are not significant enough to support their hypothesis. The methylation results in Fig. 1d, Fig. 3d, Fig. 4e, and Fig. 5f are misleading since the DNA methylation only occurs in several clones (Fig. S3, 7, 10, and 13). The DNA methylation pattern is also weird. Furthermore, subclones from PCR should be removed during bisulfite sequencing (e.g. Fig. S13).
2. Since β C specifically interact with NbROS1L, but not other member of the family, the authors should specifically silencing NbROS1L by VIGS in Fig.1.
3. The authors should use DME knock down (RNAi) mutant in *Arabidopsis* to test the viral virulence to further support their hypothesis.
4. So far, the authors still can not fully exclude the possibility that active DNA demethylation promote viral virulence through targeting the endogenous genes. In the same, β C may interact with other unknown components to enhance viral virulence.
5. Line 182, produce β - and β,δ -elimination products.
6. Line 221, transposon silencing should be transposon activation.

Reviewer #3 (Remarks to the Author):

Gui et al., 2021

The manuscript presented by Gui and colleagues describe that a geminiviral protein (β C1 from TYLCCNB, a DNA satellite associated to tomato yellow leaf curl China virus, TYLCCNV) interacts with the host DNA glycosylases, NbROS1L and AtDME and increase the demethylation activity of the latter in vitro and in vivo. The authors showed that active DNA demethylation is important for the virulence of TYLCCNV/TYLCCNB and for the level of DNA methylation at the IR. The presence of the satellite DNA induces viral accumulation and reduces the level of viral DNA methylation, whereas these observations disappear if the plant's DME-like (DMLs) DNA glycosylases are silenced. The manuscript shows convincing data for the interaction of β C1 with the N-terminal of both DNA demethylases, NbROS1 and AtDME, by two different approaches (BiFC and Co-IP) and demonstrate that Valine 17 from β C1 is essential for these interactions. The decrease in viral DNA methylation induced by the DNA satellite or the presence of β C1 is lost in a β C1V17A mutant (DNA methylation comes to levels similar to the when β C1 is not present). Moreover, the authors showed that β C1 promotes AtDME activity both in vitro and in vivo at the viral genome, and consequently, plants that overexpress AtDME and contain the tandem TYLCCNV/TYLCCNB accumulate more viral (and satellite) DNA and its genome nearly losses all DNA methylation. The findings of this work, which I believe are strongly supported by the way the data is presented, are relevant and interesting for the scientific community. Therefore, I feel that the manuscript is suitable for publication in Nature Communications but the following concerns should be addressed before publication:

1) The introduction is a bit short and significant information is missing, so it should be improved.

- Authors should include the findings from Stroud et al., 2013 (Cell) (which surprisingly is not mentioned in the text) and explain more precisely how DNA methylation is established and maintained. In the actual version, just a few lines are used to explain this complex process (lines 37-42)

- Is reference 13 correctly used in line 51? To my knowledge Lister et al., 2008 (Cell) or Stroud et al., 2013 (Cell) will be more appropriate.

- line 54: "central cells" of? Explain better

- Although the word "geminivirus" is in the title of the paper, it is not mentioned in the introduction. The authors talk about "plant viruses" and begomoviruses and give many references related to geminivirus but the section in the introduction about geminivirus should be completed and improved. Moreover, a paragraph indicating the multifunctionality of β C1 protein should be included

2) Material and Methods: The details about how VIGS (figure 1) is performed are not shown. Please indicate if TRV and the geminivirus were co-agroinoculated, at what OD, at what dpi were the samples analysed etc.

3) Figure 1b suppl.: Is DML6 expressed in seeds? It is not observed at cycle 30. Please show a better image or mention in the text that this transcript is not detected

- 4) From the information in Table1-Supplemental, I understand that the TRV-NbDMLs used in Figure 1 contains a fragment that is able to silence the six DME-like (DMLs) DNA glycosylases from *Nicotiana benthamiana*. If this is the case, it should be specifically indicated in the text (line 85). Authors should include in Figure 2a-supplementary an alignment of the fragment cloned in TRV and show and the putative sRNA hits in the six genes according to VIGS tools from Solgenomics (screen capture will be enough).
- 5) The statement “TYLCCNB promotes TYLCCNV infection...” (lines 100-101) has already been demonstrated. Please rewrite this paragraph and indicate the reference.
- 6) Authors should show that the constructs that do not give a positive interaction in Figures 2a and Fig 4b-suppl. are expressed (Western should be the easiest).
- 7) Authors should show that the C-terminal of NbROS1L in figure 2b is expressed (Western should be the easiest).
- 8) What is the rationale/criteria behind the deletions performed in BC1? Any domains? Please include a sentence in line 134 which gives more information about this to the reader.
- 9) β C1 and β C1V17A are expressed in the transgenic lines as an effect on viral levels and DNA methylation levels are observed in those plants. However, the level of expression of both transgenes should be shown (line 163) either by RT-qPCR using the same primers pair for both transgenes and/or a Western using an anti-flag antibody.
- 10) β C1 interacts with NbROS1L and AtDME through their N-terminal domain and the interaction between β C1 and AtDME and NbROS1L is lost in the β C1V17A mutant. Therefore, the authors should mention in the discussion that the lack of interaction between β C1V17A and At ROS1, AtDML2 and/or At DML3 could also be involved in the results obtained in figure 4.
- 11) Figure 4c: horizontal lines for Mock and BSCTV are misplaced. Please adjust them properly to the sample names.
- 12) The statement on line 207 is not accurate: “overexpression of DME enhances TYLCCNV infection” as the presence of β C1 or TYLCCNB is also needed. Please rephrase and explain rigorously.
- 13) The authors showed that β C1V17A mutant is able to interact with SAHH but is this mutant able to suppress gene silencing? Are there any other functions of β C1 affected in this mutant? Authors should mention this in the discussion
- 14) Authors do not show if the interaction between β C1 and AtDME or NbROS1 have any impact on the host genes controlled by these demethylases. Therefore, the verb “hijack” in the tittle seems too pretentious to me. As the authors have not shown that the interaction between BC1 and DME alters its activity in the host genome, they should rephrase the tittle of the manuscript.
- 15) The authors abuse of the use of “short sentences” throughout the manuscript that do not help the readers to follow the rationale behind some of the statements or experiments. Here are some examples but the authors should review the whole manuscript and make longer comprehensive sentences.

- lines 54-55

- lines 185-186

- lines 223-224

16) The manuscript is well-written but there are some grammar mistakes and some informal expressions not adequate for a scientific report. Authors should revise the manuscript and modify them. Here are some examples:

- line 89-90: cause disease

- line 107: represented by an intergenic region

- line 160: and the V17 residue of BC1

- lines 186: were prepared (vague)

- line 197 and 199: alone

- line 240 and 241: enhances DME-mediated DNA methylation

- line 241: not well expressed

We appreciate the constructive comments made by the reviewers. We have provided additional data and revised our manuscript to address the concerns raised by the reviewers. The revisions are highlighted. We wish the revisions are sufficient and the manuscript is now acceptable for publication. Point-by-point responses are listed below.

Reviewer #1

In this paper, Xiaojian Gui and colleagues provide compelling evidence that active viral demethylation is critical to promote virulence in coinfections by geminiviruses and satellite DNA. They show that the β C1 protein of the satellite TYLCCNB interacts with ROS1L and DEMETER (DME) glycosylases in *Nicotiana benthamiana* and *Arabidopsis thaliana*, respectively, to decrease viral DNA methylation and promote viral accumulation and virulence. This is a very nice, elegant and convincing paper. Very well done and very well written. I don't see any flaws and the data is of great significance in the field. I only have a few minor comments to do.

1) L76. In these experiments we can see the effect of TYLCCNB on TYLCCNV-infected plants that have been previously infiltrated with TRV-GUS or TRV-NbDMLs. I miss however a TRV-free control for comparison (as shown in Fig. 3). This control could be important as it has been shown that TRV interferes with DNA de/methylation (Diezma-Navas et al. Mol Plant Pathol. 2019 Oct;20(10):1439-1452. doi: 10.1111/mpp.12850. Epub 2019 Jul 5. PMID: 31274236). Is there any possible scenario by which TRV itself could influence the outcome of subsequent infections in this experiment?

Response: We thank the reviewer for the comment. We have added the TRV-free control (inoculated with pCambia1300) in Fig. 1 and Supplementary Fig. 3. The symptoms and molecular phenotypes developed by TRV-GUS-inoculated plants were exactly the same as those developed by pCambia1300-inoculated

plants, suggesting that TRV itself does not influence the outcome of TYLCCNV and TYLCCNA+B infections. Our results are consistent with previous findings (Li et al., 2014). Diezma-Navas et al found that the up-regulation of *ROS1* could be observed 7 days after TRV infection (Diezma-Navas et al., 2019). However, our results showed that the expression levels of DNA demethylases, including *NbROS1*, were comparable between TRV-*GUS*- and mock-infiltrated plants (Supplementary Fig. 3a). We speculated that the difference could result from the difference in the plant tissues used. Systemically infected leaves were used in our experiment, while inflorescences were harvested for gene expression analyses by Diezma-Navas and colleagues.

2) L124. Without a proper TYLCCNV-infected control (this control is lacking) they cannot discard a transcriptional activation of *NbDML* in TYLCCNV-infected plants, and that *NbDML* activation be further modulated when TYLCCNB is co-infecting.

Response: We agree with the reviewer's concern and TYLCCNV-inoculated control has been added in the experiment (Supplementary Fig. 4a).

3) Interestingly, viral accumulation in TYLCCNV+B-infected plants remains much higher than controls (TYLCCNV-infected plants) when β C1 carries the V17A mutation (Fig. 3C) or when DMLs are silenced (Fig. 1C). It seems likely that *ROS1L* and β C1 partially retain their functions in both cases. However, methylation levels are comparable to that observed in TYLCCNV-infected controls. Is it possible that additional mechanisms (perhaps involving other DMLs) may assist β C1 in promoting virulence?

Response: We thank the reviewer for the comment. As shown in the original figures, the methylation (especially CHH methylation) levels in TYLCCNV+B-infected plants were reproducibly, albeit mildly, lower than TYLCCNV-infected plants when β C1 carries the V17A mutation (Fig. 3d) or when *NbDMLs* are silenced (Fig. 1d). We have repeated the experiments three more times and the

new data confirmed previous results (new Fig. 1d and Fig. 3d). Considering that β C1 can target SAHH to suppress DNA methylation, we speculate that this is because SAHH activity is suppressed in TYLCCNV+B^{V17A}-inoculated plants or TRV-*NbDMLs*/TYLCCNV+B-inoculated plants, but not in TYLCCNV-inoculated plants. Our results suggest that β C1-mediated activation of NbROS1L activity and suppression of SAHH activity both contribute to β C1-mediated suppression of DNA methylation.

TYLCCNV DNA methylation is lower in TYLCCNV+B^{V17A}-inoculated plants or TRV-*NbDMLs*/TYLCCNV+B-inoculated plants than in TYLCCNV-inoculated plants. This could lead to the results that viral accumulation remains higher in TYLCCNV+B^{V17A}-inoculated plants or TRV-*NbDMLs*/TYLCCNV+B-inoculated plants than in TYLCCNV-inoculated plants (Fig. 1b, c, Fig. 3b, c). Considering that β C1 is a multifunctional virulence factor that can target different pathways to subvert host defense, we speculate that other factors may also assist β C1 in promoting virulence.

4) Viral DNA methylation in the presence of β C1(V17A) is similar to that observed in the TYLCCNV control. This mutation does not compromise the interaction between β C1 and SAHH. This seems to suggest that β C1-mediated suppression of DNA methylation via SAHH interaction is negligible (only CHH methylation is slightly reduced). However, is it possible that suppression of DNA methylation by SAHH and active demethylation by ROS1L/DME are coupled in a way that the reduced methylation observed TYLCCNV/BSCTV+B-infected plants is a consequence of both activities?

Response: Please see our response to this reviewer's comment #3.

5) Please, the 35S::RFP-H2B transgenic *N. benthamiana* plants are not mentioned in the main text. Please, describe briefly and indicate what they were used for. In general, details provided in the methods section are insufficient to reproduce the experiments.

Response: The *35S::RFP-H2B* transgenic *N. benthamiana* plants were used to mark the nucleus. We only showed the YFP channel and all the BiFC data will be uploaded as source data. We have provided more details in the methods.

Reviewer #2

This is review of the manuscript “Geminiviruses hijack host DNA glycosylases to subvert DNA methylation-mediated defense” submitted by Gui and colleagues to Nature Communications. In this study, the authors report that active DNA demethylation can target viral DNAs to promote their virulence. They also found that β C protein could directly interact with NbROS1L and DME from Arabidopsis, and enhanced the DNA glycosylase activity in vitro. In addition, they demonstrated that the protein interaction between β C and DME was essential for promote viral virulence. This study suggested a novel function of active DNA demethylation in plants. However, the evidence to support their hypothesis is not enough so far.

Specific comments:

1. The DNA methylation changes are not significant enough to support their hypothesis. The methylation results in Fig. 1d, Fig. 3d, Fig. 4e, and Fig. 5f are misleading since the DNA methylation only occurs in several clones (Fig. S3, 7, 10, and 13). The DNA methylation pattern is also weird. Furthermore, subclones from PCR should be removed during bisulfite sequencing (e.g. Fig. S13).

Response: We thank the reviewer for the comment. To address the concern that the percent of methylated cytosines is affected by the number of clones that are sequenced, we amplified the IR regions of TYLCCNV or BSCTV after bisulfite treatment and the PCR products were subjected to library construction for Illumina sequencing. Three biological replicates were performed. For data analysis, subclones from PCR were removed. We have provided the new results in the revised manuscript. The new bisulfite sequencing dataset generated in this study can be found in the NCBI Gene Expression Omnibus

under accession number GSE188968. The link is <https://www.ncbi.nlm.nih.gov/geo/query/acc.cgi?acc=GSE188968>. The token is uHQbccgsjnovpwx. The new results were similar to what was obtained before, supporting our previous conclusions (Fig. 1d, Fig. 3d, Fig. 4e and Fig. 5f). We agree with the reviewer that the CG, CHG, and CHH methylation levels of TYLCCNV IR regions do not follow the general methylation pattern in the plant genome. However, our previous and new results showed a pattern that the levels of CG, CHG and CHH methylation are comparable in the IR regions of TYLCCNV or BSCTV, which are similar to the pattern reported by other groups (Raja et al., 2008; Wang et al., 2020; Wang et al., 2019).

2. Since β C specifically interact with NbROS1L, but not other member of the family, the authors should specifically silencing NbROS1L by VIGS in Fig.1.

Response: We thank the reviewer for the suggestion. However, the sequences of *NbROS1L* and other members of the family are highly similar, especially high sequence similarity between *NbROS1L* and *NbROS1* (Supplementary Fig. 2a), making it very difficult to silence *NbROS1L* only.

3. The authors should use DME knock down (RNAi) mutant in *Arabidopsis* to test the viral virulence to further support their hypothesis.

Response: We agree with the reviewer that using DME knock down (RNAi) mutant in *Arabidopsis* to test viral virulence can further support our hypothesis. We have tried several strategies to knock down *DME* in *Arabidopsis*. However, we have not yet obtain RNAi lines in which *DME* is efficiently knocked down, very likely because *DME* is essential for endosperm development (Choi et al., 2002).

4. So far, the authors still cannot fully exclude the possibility that active DNA demethylation promote viral virulence through targeting the endogenous genes.

In the same, β C1 may interact with other unknown components to enhance vial virulence.

Response: We agree with the reviewer that we cannot exclude the possibility that active DNA demethylation could also target endogenous genes to promote vial virulence and that β C1 may interact with other unknown components to enhance vial virulence. Our study provides strong evidence that NbROS1L could be employed to promote virulence through demethylating TYLCCNV IR DNA region in the presence of β C1. We have observed that *NbDMLs* knockdown leads to consistent decreased virulence accompanied by increased DNA methylation of TYLCCNV, that overexpression of DME leads to increased virulence accompanied by decreased TYLCCNV DNA methylation, that β C1 interacts with NbROS1L as well as DME, and that β C1 promotes the demethylase activity of DME. All of these point to the conclusion that DML could be employed by the geminivirus to facilitate the infection. Furthermore, the requirement of β C1 on DML virulent function can be supported by the loss of interaction between NbROS1L and β C1^{V17A}, the absence of NbROS1L-mediated virulence change and DNA methylation changes with inoculation of TYLCCNV only, the strongly attenuated NbROS1L-mediated virulence change and DNA methylation changes accompanied by mutated β C1^{V17A} when compared to wild-type β C1.

To determine how many endogenous genes were affected when β C1 was expressed, we performed whole-genome bisulfite sequencing and compared the DNA methylation profiles of Col-0 and β C1-expressing plants. We identified 1140 hypo-differentially-methylated regions (hypo-DMRs) in β C1-expressing plants (Fig.1a in this response). Distribution analysis of these hypo-DMRs revealed that they were highly enriched in CDS, intron and intergenic region (Fig.1b in this response). GO analysis of hypo-DMR-associated genes showed that biological processes associated with these genes were not relevant to viral virulence (Fig.1c in this response), suggesting that β C1 and active DNA demethylation promote vial virulence less likely through targeting many

endogenous genes. However, it is still possible that hypomethylation of some genes contributes to β C1-mediated promotion of viral virulence and further research is needed to investigate this possibility. We have added some discussion about this in the revised manuscript and will investigate this possibility in another study.

β C1^{V17A} significantly compromised the reduction of viral DNA methylation as compared to β C1 (Fig. 3c, Fig. 4e), suggesting that the interaction between β C1 and NbROS1L or DME contributed much to the reduction of viral DNA methylation and the increase in viral virulence. As we discussed in response to Reviewer #1's comments #3 and #4, β C1 may also suppress SAHH activity to reduce viral DNA methylation and increase viral virulence.

Figure 1. **a**, Heatmap showing total methylation levels of β C1-induced hypo-DMRs in Col-0 and β C1-expressing plants. **b**, Percentages of β C1-induced hypo-DMRs derived from different genomic regions. **c**, GO analysis of genes associated with β C1-induced hypo-DMRs.

5. Line 182, produce β - and β,δ -elimination products.

Response: We thank the reviewer for pointing out this mistake, which has been corrected.

6. Line 221, transposon silencing should be transposon activation.

Response: We thank the reviewer for pointing out this mistake, which has been corrected.

Reviewer #3

Gui et al., 2021

The manuscript presented by Gui and colleagues describe that a geminiviral protein (β C1 from TYLCCNB, a DNA satellite associated to tomato yellow leaf curl China virus, TYLCCNV) interacts with the host DNA glycosylases, NbROS1L and AtDME and increase the demethylation activity of the latter in vitro and in vivo. The authors showed that active DNA demethylation is important for the virulence of TYLCCNV/TYLCCNB and for the level of DNA methylation at the IR. The presence of the satellite DNA induces viral accumulation and reduces the level of viral DNA methylation, whereas these observations disappear if the plant's DME-like (DMLs) DNA glycosylases are silenced. The manuscript shows convincing data for the interaction of β C1 with the N-terminal of both DNA demethylases, NbROS1 and AtDME, by two different approaches (BiFC and Co-IP) and demonstrate that Valine 17 from β C1 is essential for these interactions. The decrease in viral DNA methylation induced by the DNA satellite or the presence of β C1 is lost in a β C1V17A mutant (DNA methylation comes to levels similar to the when β C1 is not present). Moreover, the authors showed that β C1 promotes AtDME activity both in vitro and in vivo at the viral genome, and consequently, plants that overexpress AtDME and contain the tandem TYLCCNV/TYLCCNB accumulate more viral (and satellite) DNA and its genome nearly losses all DNA methylation. The findings of this work, which I believe are strongly supported by the way the data is presented, are relevant and interesting for the scientific community. Therefore, I feel that the manuscript is suitable for publication in Nature

Communications but the following concerns should be addressed before publication:

1) The introduction is a bit short and significant information is missing, so it should be improved.

- Authors should include the findings from Stroud et al., 2013 (Cell) (which surprisingly is not mentioned in the text) and explain more precisely how DNA methylation is established and maintained. In the actual version, just a few lines are used to explain this complex process (lines 37-42)

Response: We thank the reviewer for the comment. We have included the findings from Stroud et al., 2013 (Cell). As our study is focused on the role of active DNA demethylation in regulating geminiviral virulence, we would like to keep our introduction about DNA methylation concise.

- Is reference 13 correctly used in line 51? To my knowledge Lister et al., 2008 (Cell) or Stroud et al., 2013 (Cell) will be more appropriate.

Response: We have added the references Lister et al., 2008 and Stroud et al., 2013 in line 51. All of the three studies performed bisulfite sequencing using *ros1 dml2 dml3 (rdd)* mutant and revealed that ROS1, DML2 and DML3 target thousands of endogenous loci.

- line 54: "central cells" of? Explain better

Response: We have added a description of central cells.

- Although the word "geminivirus" is in the title of the paper, it is not mentioned in the introduction. The authors talk about "plant viruses" and begomoviruses and give many references related to geminivirus but the section in the introduction about geminivirus should be completed and improved. Moreover, a paragraph indicating the multifunctionality of β C1 protein should be included

Response: We have added sentences to introduce geminiviruses and multifunctionality of β C1.

2) Material and Methods: The details about how VIGS (figure 1) is performed are not shown. Please indicate if TRV and the geminivirus were co-agroinoculated, at what OD, at what dpi were the samples analysed etc.

Response: We thank the reviewer for the suggestion. The information has been added in the “virus inoculation” section of Material and Methods

3) Figure 1b suppl.: Is DML6 expressed in seeds? It is not observed at cycle 30. Please show a better image or mention in the text that this transcript is not detected

Response: We did not detect *NbDML6* in seeds, and this has been mentioned in the text.

4) From the information in Table1-Supplemental, I understand that the TRV-NbDMLs used in Figure 1 contains a fragment that is able to silence the six DME-like (DMLs) DNA glycosylases from *Nicotiana benthamiana*. If this is the case, it should be specifically indicated in the text (line 85). Authors should include in Figure 2a-supplementary an alignment of the fragment cloned in TRV and show and the putative sRNA hits in the six genes according to VIGS tools from Solgenomics (screen capture will be enough).

Response: This information now has been added in Supplementary Fig. 2. To improve efficiency of TRV-mediated VIGS, three fragments of *NbROS1*, *NbDML3*, and *NbDML4* were fused by overlap PCR and cloned into pTRV2 vector.

5) The statement “TYLCCNB promotes TYLCCNV infection...” (lines 100-101) has already been demonstrated. Please rewrite this paragraph and indicate the reference.

Response: We have modified the sentence accordingly.

6) Authors should show that the constructs that do not give a positive interaction in Figures 2a and Fig 4b-suppl. are expressed (Western should be the easiest).

Response: We tried several times to detect the expression of the DNA demethylases by western blot but did not yield strong signals for full-length proteins. We thus used RT-PCR instead to show that they were expressed at comparable levels (Supplementary Fig. 4c).

7) Authors should show that the C-terminal of NbROS1L in figure 2b is expressed (Western should be the easiest).

Response: The data have been provided in Supplementary Fig. 4d.

8) What is the rationale/criteria behind the deletions performed in β C1? Any domains? Please include a sentence in line 134 which gives more information about this to the reader.

Response: 1-10 amino acids, 11-20 amino acids, 21-30 amino acids, and so on, were deleted from β C1 and the schematic representation of different truncated forms of β C1 has been shown in Supplementary Fig. 5a.

9) β C1 and β C1V17A are expressed in the transgenic lines as an effect on viral levels and DNA methylation levels are observed in those plants. However, the level of expression of both transgenes should be shown (line 163) either by RT-qPCR using the same primers pair for both transgenes and/or a Western using an anti-flag antibody.

Response: The data have been provided in Supplementary Fig. 9a in the original manuscript, and now they are shown in Supplementary Fig. 8a.

10) β C1 interacts with NbROS1L and AtDME through their N-terminal domain and the interaction between β C1 and AtDME and NbROS1L is lost in the β C1V17A mutant. Therefore, the authors should mention in the discussion that

the lack of interaction between β C1V17A and At ROS1, AtDML2 and/or At DML3 could also be involved in the results obtained in figure 4.

Response: Our BiFC results revealed that β C1 does not interact with AtROS1, AtDML2 and/or AtDML3, Thus, loss of interaction between β C1^{V17A} and AtROS1, AtDML2 and/or AtDML3 could not account for the results obtained in Fig. 4. In Fig. 4, when β C1^{V17A} was overexpressed, loss of interaction between β C1 and DME accounted for the restoration of viral DNA methylation. Viral DNA methylation only partially restored, suggesting that β C1-mediated suppression of SAHH activity contributed to β C1-mediated suppression of viral DNA methylation.

11) Figure 4c: horizontal lines for Mock and BSCTV are misplaced. Please adjust them properly to the sample names.

Response: We thank the reviewer for pointing out this, and they have been adjusted.

12) The statement on line 207 is not accurate: “overexpression of DME enhances TYLCCNV infection” as the presence of β C1 or TYLCCNB is also needed. Please rephrase and explain rigorously.

Response: We agree with the reviewer that the presence of β C1 or TYLCCNB is also needed. The second half of the sentence in the original manuscript indicated this.

13) The authors showed that β C1V17A mutant is able to interact with SAHH but is this mutant able to suppress gene silencing? Are there any other functions of β C1 affected in this mutant? Authors should mention this in the discussion

Response: Our bisulfite sequencing results showed that viral DNA methylation in TYLCCNV+B^{V17A}-inoculated plants was lower than that in TYLCCNV-inoculated plants (Fig. 3d) and viral DNA methylation in β C1^{V17A}-OX lines was

lower than that in Col-0 (Fig. 4e), suggesting that $\beta C1^{V17A}$ can still suppress DNA methylation by interacting with SAHH. We have added more discussions about this in the revised manuscript. It remains unclear whether other functions of $\beta C1$ are affected.

14) Authors do not show if the interaction between $\beta C1$ and AtDME or NbROS1 have any impact on the host genes controlled by these demethylases. Therefore, the verb “hijack” in the title seems too pretentious to me. As the authors have not shown that the interaction between BC1 and DME alters its activity in the host genome, they should rephrase the title of the manuscript.

Response: We agree with the reviewer and we have rephrased the title.

15) The authors abuse of the use of “short sentences” throughout the manuscript that do not help the readers to follow the rationale behind some of the statements or experiments. Here are some examples but the authors should review the whole manuscript and make longer comprehensive sentences.

- lines 54-55

- lines 185-186

- lines 223-224

Response: We have modified short sentences throughout the manuscript.

16) The manuscript is well-written but there are some grammar mistakes and some informal expressions not adequate for a scientific report. Authors should revise the manuscript and modify them. Here are some examples:

- line 89-90: cause disease

- line 107: represented by an intergenic region

- line 160: and the V17 residue of BC1

- lines 186: were prepared (vague)

- line 197 and 199: alone

- line 240 and 241: enhances DME-mediated DNA methylation
- line 241: not well expressed

Response: We have corrected the grammar mistakes.

References:

- Choi, Y., Gehring, M., Johnson, L., Hannon, M., Harada, J.J., Goldberg, R.B., Jacobsen, S.E., and Fischer, R.L. (2002). DEMETER, a DNA glycosylase domain protein, is required for endosperm gene imprinting and seed viability in *Arabidopsis*. *Cell* 110, 33-42.
- Diezma-Navas, L., Perez-Gonzalez, A., Artaza, H., Alonso, L., Caro, E., Llave, C., and Ruiz-Ferrer, V. (2019). Crosstalk between epigenetic silencing and infection by tobacco rattle virus in *Arabidopsis*. *Mol Plant Pathol* 20, 1439-1452.
- Li, F., Huang, C., Li, Z., and Zhou, X. (2014). Suppression of RNA silencing by a plant DNA virus satellite requires a host calmodulin-like protein to repress RDR6 expression. *PLoS Pathog* 10, e1003921.
- Raja, P., Sanville, B.C., Buchmann, R.C., and Bisaro, D.M. (2008). Viral genome methylation as an epigenetic defense against geminiviruses. *J Virol* 82, 8997-9007.
- Wang, L., Ding, Y., He, L., Zhang, G., Zhu, J.K., and Lozano-Duran, R. (2020). A virus-encoded protein suppresses methylation of the viral genome through its interaction with AGO4 in the Cajal body. *Elife* 9.
- Wang, Y., Wu, Y., Gong, Q., Ismayil, A., Yuan, Y., Lian, B., Jia, Q., Han, M., Deng, H., Hong, Y., *et al.* (2019). Geminiviral V2 protein suppresses transcriptional gene silencing through interaction with AGO4. *J Virol* 93.

REVIEWERS' COMMENTS

Reviewer #1 (Remarks to the Author):

This a second review of a previously submitted manuscript by Xiaojian Gui and colleagues entitled Geminiviruses employ host DNA glycosylases to subvert DNA

methylation-mediated defense. I was very much enthusiastic about the first version of this paper, but I agree that the comments made by the reviewers and the manner these comments have been addressed in this revised version have contributed to improve greatly the overall quality of the work. Particularly, I am very much satisfied with their response to my comments and the correction made when necessary. I feel they have also provided solid arguments in response to the other reviewers' concerns. Again, I don't see any relevant flaws in the data analysis and interpretation, and the conclusions are fully supported by their results. Now, the methodology is comprehensive. In conclusion, I fully support this paper for publication.

Reviewer #2 (Remarks to the Author):

The authors have successfully addressed all of my concerns.

Reviewer #3 (Remarks to the Author):

The manuscript has clearly been improved after the changes introduced by the authors, following the reviewers' comments. I just have a minor comment from a new sentence that the authors have included in the final version of the introduction:

Line 66-67: interfere with the RNA silencing pathway (REF 22, 23)

As β C1 can suppress DNA methylation through inactivating SAHH (REF 22) but not by interfering with the RNAi machinery that leads to DNA methylation, I believe the sentence in line 66-67 should be corrected as follows: interfere with post-transcriptional and transcriptional gene silencing (REF 23, 22)

In my opinion, this final form of the manuscript could be published in Nature Communications.

We appreciate the constructive comments made by the reviewers. We wish the revisions are sufficient and the manuscript is now acceptable for publication. Point-by-point responses are listed below.

Reviewer #1

This a second review of a previously submitted manuscript by Xiaojian Gui and colleagues entitled Geminiviruses employ host DNA glycosylases to subvert DNA methylation-mediated defense. I was very much enthusiastic about the first version of this paper, but I agree that the comments made by the reviewers and the manner these comments have been addressed in this revised version have contributed to improve greatly the overall quality of the work. Particularly, I am very much satisfied with their response to my comments and the correction made when necessary. I feel they have also provided solid arguments in response to the other reviewers' concerns. Again, I don't see any relevant flaws in the data analysis and interpretation, and the conclusions are fully supported by their results. Now, the methodology is comprehensive. In conclusion, I fully support this paper for publication.

Response: We thank the reviewer for his/her positive assessment of our manuscript.

Reviewer #2

The authors have successfully addressed all of my concerns.

Response: We thank the reviewer for his/her positive assessment of our manuscript.

Reviewer #3

The manuscript has clearly been improved after the changes introduced by the authors, following the reviewers' comments. I just have a minor comment from a new sentence that the authors have included in the final version of the introduction:

Line 66-67: interfere with the RNA silencing pathway (REF 22, 23)

As β C1 can suppress DNA methylation through inactivating SAHH (REF 22) but not by interfering with the RNAi machinery that leads to DNA methylation, I believe the sentence in line 66-67 should be corrected as follows: interfere with post-transcriptional and transcriptional gene silencing (REF 23, 22)

In my opinion, this final form of the manuscript could be published in Nature Communications

Response: We thank the reviewer for his/her positive assessment of our manuscript. We agree with the reviewer and we have changed the sentence in line 66-67 as follows: The multifunctional protein β C1 can block MAPK signaling, interfere with post-transcriptional and transcriptional gene silencing and manipulate jasmonic acid signaling to subvert host defense.